

# Potential role of *Manilkara Zapota L* in treating bacterial infection

Ami Febriza[1,2], Fityatun Usman[1], Andi Ulfah Magefirah Rasyid[1],
Hasta Handayani Idrus[2] and Mohd Helmy Mokhtar[3]

[1] Faculty of Medicine and Health Sciences, Universitas Muhammadiyah Makassar, Makassar, South Sulawesi, Indonesia
[2] Center for Biomedical Research, Research Organization for Health, National Research and Innovation Agency (BRIN), Cibinong Bogor, West Java, Indonesia
[3] Departement of Physiology, Universiti Kebangsaan Malaysia, Kuala Lumpur, Malaysia

## ABSTRACT

The increasing problem of antibiotic resistance in bacteria leads to an urgent need for new antimicrobial agents. Alternative treatments for bacterial infections need to be explored to tackle this issue. Plant-based substances are emerging as promising options. *Manilkara zapota* L. contains compounds with antibiotic activities, and anti-inflammatory, antitumor, antipyretic, and antioxidant properties. It has medicinal properties and contains bioactive compounds, like tannins, flavonoids, and triterpenoids. This review aimed to comprehensively evaluate the existing literature on the potential medicinal and therapeutic benefits of *M. zapota* in bacterial infections by utilizing data from *in vivo* and *in vitro* studies. *M. zapota* has the potential to be a nutritional source of antimicrobial food. Numerous preclinical studies have demonstrated the antibacterial activities of *M. zapota* and its components. The antibacterial mechanisms of this fruit could interact with bacterial cell structures such as cell walls or membranes.

# INTRODUCTION

Antibiotic resistance is a growing concern, highlighting the need for new antimicrobial agents (*Russell, 2002*). Over the past few decades, the surge in drug resistance among Gram-positive bacteria has been attributed to bacterial evolution and the rampant overuse/misuse of antibiotics. This surge has intensified global antibiotic resistance, complicating clinical treatment (*Guo et al., 2020*). Additionally, there is a growing incidence of infections caused by multidrug-resistant Gram-negative bacteria, contributing to elevated morbidity and mortality rates and prolonged hospitalization (*Cerceo et al., 2016*).

Addressing antibiotic resistance to clinically essential pathogens involves exploring new treatment options for bacterial infections using plant-based substances emerging as promising alternatives. Studies have indicated that plants harbor antimicrobial compounds capable of augmenting or diminishing antibiotic activity (*Chandra et al., 2017*; *Fankam, Kuiate & Kuete, 2017*). Natural products such as herbs provide unique molecular diversity and biological functionality, rendering them valuable for drug discovery (*Gootz, 1990*).

Corresponding author
Ami Febriza,
amifebriza@med.unismuh.ac.id

*Manilkara zapota* L., a member of the Sapotaceae family, commonly known as sapodilla, produces milky juice; it is also known by various names, including *Manilkara zapotilla*, *Mimusops manilkara*, *Achras zapota*, and *Achras sapota* (*Lim, 2013*). Extensive evidence supports the medicinal properties of this plant, including its antimicrobial activity (*Idrus et al., 2019*), anti-inflammatory, antipyretic (*Ganguly et al., 2013*), antitumor (*Khalek et al., 2015*), and antioxidant properties (*Islam et al., 2011*). Bioactive compounds found in *M. zapota* include tannins, flavonoids, and triterpenoids (*Fayek et al., 2013*). The leaves and seeds also contain triterpenes, tannins, and polyphenols, which have been investigated for their potential medicinal properties (*Ngongang et al., 2020*).

### Rationale of the study

The medicinal potential of *Manilkara zapota* L. is reviewed in relation to combating antibiotic resistance. This plant contains bioactive compounds that have been researched for their antimicrobial properties (*Idrus et al., 2019*), which can be found in various parts of the plant. These bioactive compounds could provide a natural source for developing new treatment options to fight antibiotic-resistant bacteria. The potential for plant-based substances to address antibiotic resistance is a promising area for discovering new antimicrobial agents. This study aimed to comprehensively review and evaluate the existing literature on the potential medicinal and therapeutic benefits of *M. zapota* against bacterial infections. This review includes data from both *in vivo* and *in vitro* animal studies.

## SURVEY METHODOLOGY

A literature search was performed to identify and present relevant articles on the effects of *Manilkara zapota* L on bacterial infections. Electronic databases such as Pubmed, Scopus, and Google Scholar were searched for peer-reviewed full-text articles to identify articles published in English and covering the period from 2013 to 2023. The following keywords were used in the search: (1) "*Manilkara Zapota*" or "*M. Zapota*" or "*M. Zapota* Extract," (2) "Bacterial Infection," and (3) "Antimicrobial," "Antioxidant," or "Anti-inflammatory." All studies, encompassing both *in vitro* and *in vivo* studies, examined the effects of *Manilkara zapota* L on infections and focused on its antimicrobial properties. The identified impacts included both quantitative and qualitative evaluations. The quantitative element included the determination of inhibition zones or the reduction of biomarkers. In contrast, the qualitative aspect included evaluating of the antibacterial activities of *M. zapota* and the chemical or natural substances. Studies on M. *zapota* that did not delve into infection-related issues or bioactive properties such as antimicrobial, antioxidant, or anti-inflammatory effects were excluded from the review.

Figure 1 shows the flowchart of article selection. The identification in the database resulted in 2,520 articles, which were reduced to 35 after screening. After the duplicate articles were identified, each was screened according to the exclusion criteria, so that finally 15 full articles reporting *in vivo* and *in vitro* studies were screened for eligibility.
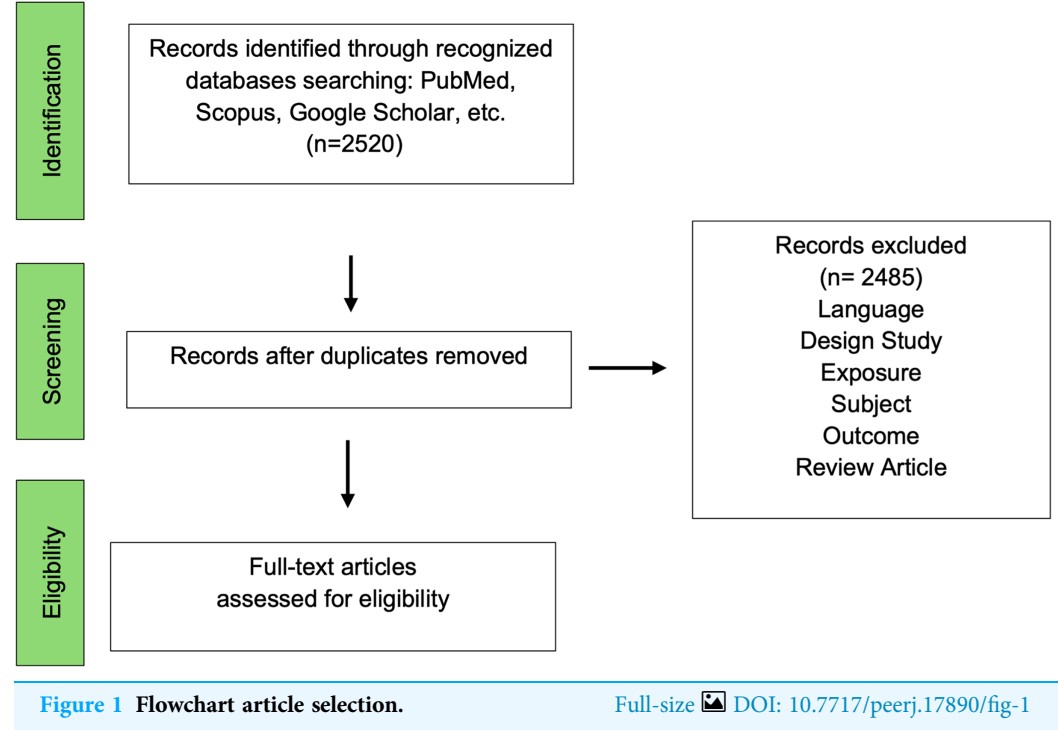

**Figure 1 Flowchart article selection.**

## Botanical description of M. *zapota*

The sapodilla fruit, a nearly round brown berry with a width ranging from 5 to 10 cm, transforms its texture as it ripens. When unripe, the fruit is hard and coarse. However, as it ripens, it becomes soft and juicy. It can be either round or egg-shaped, weighing between 75 and 200 g (*Tulloch et al., 2020*). The M. *zapota* plants are shown in Fig. 2. The fruit pulp is light brown, soft, easily digestible, and has a gritty texture; it harbors 3–12 black seeds containing phytochemicals such as saponins (*Ahmed et al., 2008*). Sapodilla trees have a shallow root system with most roots situated within the top 75 cm of the soil. Approximately 66% of the moisture extracted from the soil is concentrated in the upper layers. The leaves of this evergreen tree are arranged in a spiral and measure 7–12 to 2–4 cm. Initially pinkish-brown when young, they transition to a light green to dark-green hue as they mature (*Mehnaz & Bilal, 2017*; *Mickelbart, 1996*).

## Phytochemical composition

Various chemical compounds have been identified in *M. zapota*, and they are rich in phytochemicals and antioxidants (*Lobo et al., 2010*). However, it is essential to note that the composition of these compounds varies based on geographical location and isolation methods. The constituents of *M. zapota* include common compounds such as carbohydrates, proteins, fats, fiber, vitamins, and minerals (*Ma et al., 2003*; *Punia Bangar et al., 2022*). Additionally, this plant contains bioactive phytochemicals, such as triterpenes, saponins, tannins, and polyphenols (*Ngongang et al., 2020*). Table 1 lists the chemical composition of *M. zapota*.

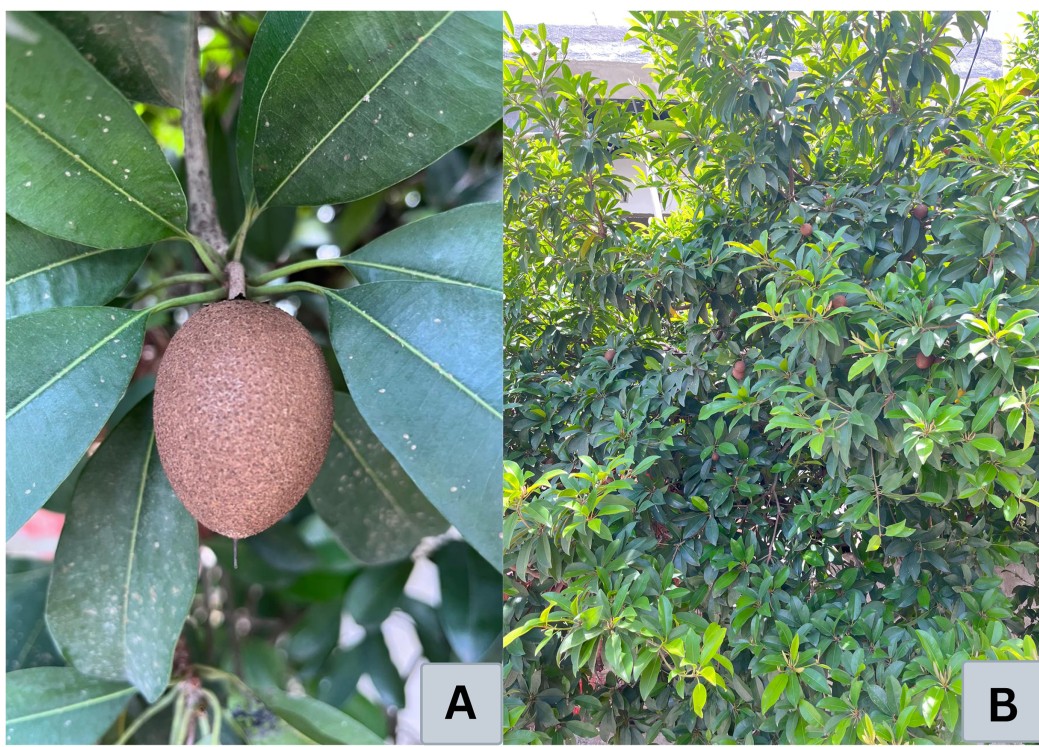

**Figure 2 (A) Fruit and (B) plant of *Manilkara zapota* L.** The M. *zapota* plants are shown in Fig. 2. (A) M. *zapota* fruit has a round or oval shape with light brown skin. (B) M. *zapota* has evergreen trees with dark green, leathery leaves (source: documentation of Fityatun Usman).

**Table 1 List of the chemical composition of *Manilkara zapota* L.**

| Component | Composition | References |
|---|---|---|
| Carbohydrates | 19.9 g | *Miranda (2022)* |
| Protein | 0.44 g | *Miranda (2022)* |
| Fat | 1.10 g | *Miranda (2022)* |
| Fibre | 5.3 g | *Miranda (2022)* |
| Vitamin | Vitamin A, Vitamin C, Folate, Niacin | *Miranda (2022)* and *Moo-Huchin et al. (2014)* |
| Minerals | Calcium, potassium, iron, copper, magnesium, zinc | *Miranda (2022)* |
| Terpenes and terpenoids | β-amyrin-3-(3′-dimethyl) butyrate | *Fayek et al. (2013)* |
| Phenolic | Chlorogenic acid, gallic acid, p-hydroxybenzoic, ellagic acid, quercetin, ferulic acid, catechin, kaempferol, trans cinnamic acid | *Ma et al. (2003)*, *Moo-Huchin et al. (2014)*, *Nile & Park (2014)* and *Singh et al., 2016*) |
| Flavonoid | Quercetin, apigenin-7-O-β-D-glucuronide methyl ester | *Kamalakararao et al. (2018)*, *Moo-Huchin et al. (2014)* and *Singh et al. (2016)* |
| Carotenoid | Lycopene | *Moo-Huchin et al. (2014)* and *Da Silva et al. (2014)* |
| Saponin | Manilkoraside | *Fernandes et al. (2013)* and *Linn et al. (2012)* |
| Tannin | Condensed tannin | *Hamzah et al. (2020)* |

The nutritional richness and potential health benefits of *M. zapota* are emphasized by its diverse range of nutrients, including vitamins, minerals, and phytochemicals. This fruit is a valuable source of essential macronutrients in small amounts, including carbohydrates, proteins, and fats, which provide energy and support various bodily functions (*Miranda, 2022*). It is also a source of important vitamins, such as A, C, folate, and niacin, as well as essential minerals, such as calcium, potassium, iron, copper, magnesium, and zinc, all of which are critical for various physiological processes (*Miranda, 2022*). A previous study found that *M. zapota* contains 21.43 mg of vitamin C, 1.16 mg of total anthocyanins, 15.35 mg of phenolic compounds, 0.18 mg of quercetin, and 1.69 mg of carotenoids per 100 g (*Moo-Huchin et al., 2014*).

One noteworthy natural compound isolated from *M. zapota* is β-amyrin-3-(3′-dimethyl) butyrate, a terpenoid identified using spectral methods such as IR, MS, UV, 1H-NMR, 13C-NMR, and 2D-NMR. Together with other compounds, including lupeol-3-acetate and 4-caffeoylquinic acid, it is found in alcoholic and aqueous extracts of unripe *M. zapota* fruits (*Fayek et al., 2013*).

*M. zapota* also contains various phenolic compounds, including chlorogenic acid, quercetin, *p*-hydroxybenzoic acid, ellagic acid, ferulic acid, gallic acid, catechin, trans cinnamic acid, and kaempferol. These phenolic compounds significantly enhance the antioxidant capacity of and contribute to the potential health benefits of *M. zapota* (*Ma et al., 2003*; *Singh et al., 2016*).

Quercetin is a phenolic compound found in various fruits, including *M. zapota* (Sapodilla). Another bioactive flavonoid compound found in the ethyl acetate leaf extract of *M. zapota* is apigenin-7-O-β-D-glucuronide methyl ester, which exhibits considerable DPPH (1,1-diphenyl-2-picrylhydrazyl) and NO (nitric oxide) free radical scavenging activity, showing its promise as an herbal antioxidant. Both quercetin and apigenin-7-O-β-D-glucuronide methyl ester contribute to the antioxidant activities of *M. zapota*, suggesting their potential for developing therapeutic antioxidants (*Singh et al., 2016*).

Carotenoids, which give fruits and vegetables bright colors. Sapodilla contains specific carotenoids such as beta-carotene and lycopene, which can be quantified using spectrophotometry (*Da Silva et al., 2014*). In *M. zapota*, the compound 3-O-β-D-glucopyranosyl-(1→6)-β-D-glucopyranosyl-28-o-α-L-rhamnopyranosyl-(1→3)-β-D-xylopyranosyl-(1→6)-(1→4)-α-L-rhamnopyranosyl-(1→2)-α-L-arabinopyranosyl-(1→3)-β-D-xylopyranosyl-(1→6)-(1→6)-β-D-xylopyranosyl-(1→6)-(1→6)-β-D-xylopyranosyl-(1→6)-(1→6)-β-D-xylopyranosyl-(1→6)-(1→6)-β-D-xylopyranosyl-(1→6) acid is present. This compound, a condensed tannin and polymer of flavonoids. Saponins, another class of plant secondary metabolites, are distributed in different organs and tissues. As antimicrobial plant-protection agents, saponins play a role in plant defense against soil pathogens (*Ahmed et al., 2008*). Manilkoraside was extracted from the stem bark of *M. zapota* with ethanol. From a previous study, 20 and 48 μg/ml of manilkorasid could suppress the growth of HL-60 and HT29 cancer cells by cytotoxicity assays measurement (*Linn et al., 2012*).

**Table 2 Phytochemical isolated from part plants of *Manilkara zapota* L.**

| Plant part | Phytochemical compounds | References |
|---|---|---|
| Leaf | myricetin-3-O-α-L-rhamnoside, 2-oleanene-3,28-diol, Chlorogenic acid, myricetin-3-O-β-D-glucopyranoside, mearnsitrin, germanicol and germanicol acetate | *Mourão Mulvaney et al. (2021)*, *Osman et al. (2014)* and *Rao et al. (2014)* |
| Bark | Stigmasterol. β-sitosterol, lupeol, lupenone, glut-5(6)-en-3β-acetate, olean-12-en-3β-acetyl-11-one | *Noor et al. (2014)* |
| Fruit | β-amyrin-3-(3′-dimethyl) butyrate, lupeol-3-acetate, 4-Caffeoylquinic acid (Cryptochlorogenic acid) | *Fayek et al. (2013)* |
| Seed | Saccharose, D-quercitol | *Rao et al. (2014)* |

In addition, natural compounds found in *M. zapota* exhibit various activities such as antimicrobial, antioxidant, antihyperglycemic, antitumor and hypocholesterolemic activity. The health benefits associated with *M. zapota* are attributed to these compounds (*Ahmed et al., 2008*; *Fayek et al., 2013*; *Linn et al., 2012*; *Da Silva et al., 2014*; *Singh et al., 2016*).

Numerous compounds have been isolated and identified from different parts of *M. zapota*, as shown in Table 2. The anticancer effect of M. *zapota* leaves has been studied and the constituent of the extract, erythrodiol, was found to play a role in reducing the number of live tumor cells (*Osman et al., 2014*). In another study examining the leaves of *M. zapota*, they were reported to contain phenolic chemicals and pentacyclic triterpenes, including chlorogenic acid, myricetin-3-O-β-D-glucopyranoside, mearnsitrin, germanicol, and germanicol acetate. These compounds showed notable efficacy against *Candida albicans* and *Staphylococcus aureus* (*Mourão Mulvaney et al., 2021*). In addition, myricetin-3-O-α-L-rhamnoside isolated from the leaves had biological activities such as antioxidant, moderate elastase inhibitory, and tyrosinase inhibitory activities (*Rao et al., 2014*). A previous study using the methanol-extracted stem bark of *M. zapota* revealed the presence of compounds including stigmasterol, β-sitosterol, lupeol, lupenone, glut-5(6)-en-3β-acetate, and olean-12-en-3β-acetyl-11-one (*Noor et al., 2014*).

Phytochemical screening of *M. zapota* fruits revealed the presence of β-amyrin-3-(3′-dimethyl) butyrate, lupeol-3-acetate, 4-Caffeoylquinic acid (cryptochlorogenic acid) (*Fayek et al., 2013*). In this prior investigation, methanol extracted from the fruit of *M. zapota* showed a significant reduction in cholesterol levels and a moderate improvement in glucose levels (*Fayek et al., 2013*). Phytochemical screening of *M. zapota* oil seeds showed the presence of functional groups in alkanes, carbonyl, alkene and methyl ester (*Mehedi et al., 2023*). In another study, saccharose and D-quercitol were reported to be isolated from *M. zapota* seeds (*Rao et al., 2014*).

## Effect of M. *zapota* found in preclinical studies
### *In vitro studies*

Previous studies have demonstrated the antibacterial properties of *M. zapota* extracts against various strains. These studies primarily assessed the effects of the extracts on bacterial growth under laboratory conditions, as summarized in Table 3.

**Table 3 Summary of *in vitro* studies of *Manilkara zapota* L. in bacterial infection.**

| Type of study | Parameter | Dose | Outcomes | References |
|---|---|---|---|---|
| The extract of peel using hexane, ethyl acetate, acetone, methanol and water | % Inhibition of microorganism | Concentration used was 20 mg/ml | Gram positive (36%), gram negative (92%), fungi (70%) | *Rakholiya, Kaneria & Sumitra Chanda (2014)* |
| Acetone extract from seed | Agar well diffusion assay; GC-MS and FT-IR methods | (67.15 ± 4.35 mg/g) of total phenolics, 49.93 ± 8.76 mg/g of tannins, and 60.06 ± 6.4 mg/g of flavonoids were found | The highest zone of inhibition was reported for MRSA E-1122, *Candida albicans*, and *Micrococcus luteus* | *Mohanapriya et al. (2019)* |
| Methanol extract of seed and leaves | Minimal inhibitory concentration (MIC) | 256 µg/mL ≤ MIC ≤ 1,024 µg/mL | Increased antibiotics activity up to 16-fold at their MIC/2 and MIC/4 | *Ngongang et al. (2020)* |
| Ethanol extract of bark and leaves | The disc diffusion method | 400 µg/disc for leaves and bark extract | Bark extract (400 µg/disc) was effective against all bacteria tested (7–13.5 mm zone of inhibition) | *Islam (2013)* |
| Hydroethanolic extract of leaves | Minimum inhibitory concentration (MIC) | The concentration of plaque ranged from 8–512 µg/mL | Antibacterial activity against the standard *Staphylococcus aureus* strain was reported using the extract | *Freitas et al. (2021)* |
| Water extract of the root | The agar cup plate method | 0.5 extract plant powder, 2.5% yield of the powdered plant | Zone inhibition for *S. aureus* (11 mm) and *E. coli* (18 mm) | *Nama et al. (2013)* |
| Methanol extract of the leaves of *M. zapota* and *P. guajava* | The well method | Five ratio of concentration extract *M. zapota* and *P. guajava* (2:0, 1, 6:0, 4, 0.6:1, 4, 1:1, 0:2) | Zone inhibition for *S. aureus* (18 mm), *S. typhi* (13.8 mm), *E. coli* (13.8 mm), *Bacillus subtilis* (20 mm) | *Salunkhe et al. (2018)* |

**Note:**
GC-MS, Gas chromatography-mass spectrometry; FT-IR, Fourier transform infrared; MRSA, Methicillin resistant *Staphylococcus aureus*; MIC, Minimum inhibitory concentration.

Previous studies have shown that *M. zapota* exhibits antibacterial properties. The ethyl acetate extract derived from the stem bark showed efficacy against all pathogenic bacteria and certain fungi. The ethyl acetate extract from the leaves exhibited modest antibacterial activity against some microorganisms, with minimum inhibitory concentrations (MICs) ranging from 256 to 512 µg/ml (*Osman et al., 2011*). Another study reported the use of bark and leaf extracts. The bark extract exhibited notable antibacterial efficacy against all tested bacteria, with an average zone of inhibition of 7 to 13.5 mm. In contrast, the leaf extract showed lower efficacy and had no activity against *Staphylococcus aureus* (*Islam, 2013*).

A study examined the biological activity of *M. zapota* seed extracts. The extract contains various of bioactive components, including phenolics, tannins, and flavonoids. The seed extract demonstrated notable antibacterial activity against human infections and multidrug-resistant methicillin-resistant *S. aureus* (MDR-MRSA) (*Mohanapriya et al., 2019*). *M. zapota* root extract exhibited MIC values of 25–100 mg/ml against *S. aureus* and *Escherichia coli* (*Nama et al., 2013*). In another previous study, the hydroethanolic extract of *Manilkara zapota* leaves (HEMZL) contained tannins and phenolic chemicals. It was effective against *Staphylococcus aureus* but had antagonistic effects or showed no significant difference when used with antibiotics (*Freitas et al., 2021*). Another study using a Sapodilla extract at a dosage of 20 mg/ml revealed promising anti-adhesion properties

that inhibited the adhesion of *H. pylori* to mucosal surfaces. The method of inhibiting adhesion to mucosal surfaces is beneficial for preventing infection and reducing bacterial resistance (*Chaichanawongsaroj & Pattiyathanee, 2014*). In another study, nanoparticles were effecient as antimicrobial agents due to their large surface area, chemical reduction characteristics, and surface reactivity (*Rai et al., 2021*). The copper nanoparticles (Mz-Cu-NPs) of *M. zapota* isolated from aqueous leaves were shown to have antibacterial activity. The nanoparticles showed significant antibacterial efficacy in comparison to the control sample. The degree of inhibition increased with higher doses, with *B. subtilis* showing minimal inhibition of up to 50% (*Kiriyanthan et al., 2020*). The results of the previous studies add to the growing body of evidence that *M. zapota* has antibacterial activity against resistant strains.

### In vivo studies

Animal studies evaluating the antibacterial properties of *M. zapota* provided insights into its effects on living animals (Table 4). In a study involving mice infected with *Salmonella typhi*, the extract from manila sapodilla fruit significantly reduced TNF-α levels. This study aimed to assess the efficacy of *M. zapota* against bacterial infections. All experimental groups exhibited a notable decrease in TNF-α concentration on days 4, 10, and 30 post-treatment. Particularly, the group treated with manila sapodilla fruit extract demonstrated a significant reduction in TNF-α concentration (*Idrus et al., 2022*). Previous studies identified various bioactive mechanisms, including antioxidant, anti-inflammatory, cytotoxic, antidiabetic, and analgesic effects (Table 4).

## Antibacterial properties/mechanisms of action

*M. zapota* has promising applications in medical and industrial settings because of its antibacterial properties, rendering it adaptable to sectors such as healthcare and food processing to combat bacterial threats (*Shahraki, Javar & Rahimi, 2023*; *Rivas-Gastelum et al., 2023*). Numerous studies have highlighted the potential of *M. zapota* as a source of therapeutic agents with antimicrobial properties tested on various plant parts, including leaves, bark, seeds, and roots (Table 2).

*M. zapota* exhibits a broad antibacterial spectrum against clinically significant Gram-positive and Gram-negative bacteria, including multidrug-resistant strains. Notably, the leaf extracts generally increased activity than the bark and seed extracts (*Cerceo et al., 2016*; *Guo et al., 2020*; *Shahraki, Javar & Rahimi, 2023*). Previous research on the stem bark and leaf extracts revealed the presence of terpenoids, glycosides, and flavonoids. A recent study demonstrated the antimicrobial properties of the extract against various bacteria including *Salmonella typhi, Sarcina lutea, Bacillus megaterium, Bacillus subtilis*, and *E. coli*, with inhibition zones ranging from 8 to 16 mm (*Osman et al., 2011*). Terpenoids and phenylpropanoids exhibit antimicrobial activity by damaging the bacterial membranes, thereby altering their permeability. This effect was confirmed by assessing salt tolerance, release of cellular constituents, and crystal violet absorption in *E. coli* and *S. aureus* (*Nogueira et al., 2021*). The flavonoid identified in *M. zapota* exhibits antibacterial activity,

**Table 4 Summary of *in vivo* studies of *Manilkara zapota* L.**

| Type of study | Treatment dosage | Duration of study | Outcomes | References |
|---|---|---|---|---|
| Ethanol extract of fruit; Salmonella infected mice | *Manila sapodilla* fruit extract (510 and 750 mg/kg BW) | 4, 10, and 30 days | Reducing levels in TNF-α concentration in all experimental groups at observation days 4, and 10 until day 30 after treatment. | *Idrus et al. (2022)* |
| Leaf extracts in ethyl acetate and methanol; albino mice; acute and subacute toxicity | Leaf extracts (50, 500, 1,500, and 2,000 mg/kg BW) | 28 and 29 days | Hematological and serum biochemical markers showed no significant changes. | *Bhattacharya et al. (2014)* |
| Ethanol extract of leaves; albino rats, antidepressant activity with forced swim test, tail suspension test | Extract of leaves (150 and 300 mg/kg BW) | 60 min before intervention activity | In the tail suspension and forced swim models of depression, extract significantly reduced immobility. | *Kiranmayi & Sai Sureshma (2022)* |
| Ethanol extract of seed; adult Wistar albino rats; Eddy hot plate method | Extract of seed (200 mg/kg BW) | At 0, 30, 60, and 90 min-points reaction times | Extract treatment resulted in less increase in reaction time compared to aspirin treatment at 30 and 60 min. | *Ramanna et al. (2023)* |
| Soxhlet extraction of bark; Wistar rats; excision wound model | Extract topical ointment (5% and 10%) | At 4, 8, 12, 16, and 20 days for wound diameter | Contraction of the wound was substantially greater ($p$ 0.001) in rats treated with MZE ointment than in the control group. | *Alsareii et al. (2023)* |
| Ethanol extract of leaves; Wistar mice, diabetes mellitus model | Extract of leaves (100 and 300 mg/kg BW) | 14 days | LDL levels differed significantly between groups treated with leaf extract (100 and 300 mg) and pioglitazone. | *Solikhah et al. (2023)* |
| Water extraction of leaves; mice with hyperuricemia model | Leaf extract (1 and 3 g/kg BW) | 28 days | After hyperuricemia induction, there was no significant difference in mean uric acid levels between groups treated with the leaf extract and ascorbic acid. | *Cervero et al. (2018)* |

**Note:**
BW, Body weight; TNF-α, Tumor necrosis factor alpha; MZE, *M. zapota* bark ethanolic extract; LDL, Low-density lipoprotein.

leading to cell lysis and disruption of the cytoplasmic membrane by increasing its permeability (*Tagousop et al., 2018*). The concentration of flavonoids was found to impact bacterial membranes, as observed in scanning electron microscopy images after the application of flavonoid glycosides to Pseudomonas (*Selim et al., 2012*).

In antimicrobial testing, the stem bark extract demonstrated activity against all tested pathogenic bacteria, whereas the leaf extract was effective against some bacterial strains. Moreover, stem bark extract showed activity against all three fungi, whereas the leaf extract showed no antifungal properties (*Osman et al., 2011*). The stem bark extract showed significant antibacterial activity against all bacteria tested, whereas the leaf extract displayed comparatively lower activity. The preliminary phytochemical screening revealed the presence of alkaloids, flavonoids, saponins, and tannins (*Osman et al., 2011*; *Islam, 2013*). This study also revealed that the root extract of *M. zapota* exhibited antibacterial activity against *S. aureus* and *E. coli*. The presence of bioactive constituents, including alkaloids, glycosides, saponins, tannins, and carboxylic acids, in the extract may contribute to its antimicrobial properties (*Nama et al., 2013*).

The fruit of *M. zapota* was extracted through maceration and analyzed for tannin content. This study confirmed the presence of tannins, specifically condensed tannins, in this fruit. The tannin content, determined using permanganometry, averaged 0.84%

(*Hamzah et al., 2020*). Tannins inhibit ruminal microorganisms' growth and protease activity, by targeting bacterial cell walls and potentially disrupting cell membranes if they are sufficiently lipophilic (*Cowan, 1999*). Previous studies have highlighted the efficacy of tannic acid in inhibiting biofilm formation by *S. aureus*, which is a major contributor to infections (*Dong et al., 2018*). *M. zapota* extract showed significant antibacterial properties due to diverse phytochemical compounds, such as flavonoids and alkaloids, which act synergistically against bacterial infections. The identification of these compounds suggests a multifaceted approach to bacterial inhibition. Comprehensive phytochemical analyses have shed light on the overall chemical composition of plants, paving the way for potential applications beyond antibacterial intervention. In addition to its antibacterial effects, *M. zapota* contains compounds with anti-inflammatory, antioxidant, and anticancer properties.

A previous study revealed that *M. zapota* seeds contain bioactive substances with antimicrobial properties (*Ahmed et al., 2008*; *Mohanapriya et al., 2019*). Examination of *M. zapota* seed extracts revealed significant secondary metabolites with antimicrobial activity, including tannins, flavonoids, and phenolics. GC-MS and FT-IR analyses confirmed the presence of various active compounds such as fatty acids, aliphatic amines, amines, alkanes, and alkenes. The extract exhibited antimicrobial activity against human pathogens, including multidrug-resistant strains (*Mohanapriya et al., 2019*). Methanolic seed extract demonstrates a notable MIC against multiple strains of *S. aureus* ranging from 100 to 512 μg/ml (*Ngongang et al., 2020*). Phytochemical analyses revealed the presence of steroids (*Ngongang et al., 2020*) and saponins (*Ahmed et al., 2008*) in seeds. Previous studies have indicated that steroidal saponins alter the cell walls of bacteria such as *Prevotella bryantii*, *Ruminobacter amylophilus*, *Streptococcus bovis*, and *Selenomonas ruminantium*, as observed using transmission electron microscopy (*Wang et al., 2000*). The continuous use of *M. zapota* in various cultures indicates its well-established reputation as a valuable natural medicinal resource. While this plant's antibacterial and antioxidant properties of this plant have been studied extensively, its traditional use has revealed a wide range of benefits. This versatility makes *M. zapota* more than just a source of specific therapeutic compounds; it is a holistic botanical remedy closely intertwined with cultural practices and traditional knowledge.

## CONCLUSIONS

In summary, *M. zapota* has the potential as a food source for antimicrobials. Numerous preclinical studies have highlighted the antimicrobial properties of *M. zapota* and its compounds. The antibacterial mechanisms of this fruit involve potential interactions with bacterial cell structures such as cell walls or membranes. Moreover, animal models treated with *M. zapota* extract revealed antimicrobial effects and other bioactive mechanisms. These include antioxidant, anti-inflammatory, cytotoxic, anti-diabetic, and analgesic effects. The multifaceted properties of this plant, particularly its antimicrobial properties, make it a noteworthy candidate for medicinal applications to combat infections. However, a comprehensive understanding of the mechanisms of action requires further research.

### Funding

The authors received no funding for this work.

### Competing Interests

Mohd Helmy Mokhtar is an Academic Editor for PeerJ.

### Author Contributions

- Ami Febriza conceived and designed the experiments, analyzed the data, authored or reviewed drafts of the article, and approved the final draft.
- Fityatun Usman conceived and designed the experiments, performed the experiments, prepared figures and/or tables, and approved the final draft.
- Andi Ulfah Magefirah Rasyid conceived and designed the experiments, prepared figures and/or tables, and approved the final draft.
- Hasta Handayani Idrus analyzed the data, authored or reviewed drafts of the article, and approved the final draft.
- Mohd Helmy Mokhtar performed the experiments, analyzed the data, authored or reviewed drafts of the article, and approved the final draft.

### Data Availability

This is a literature review, hence it did not utilize raw data/code.

### Supplemental Information

Supplemental information for this article can be found online at http://dx.doi.org/10.7717/peerj.17890#supplemental-information.

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
