# Peer review of "Potential role of Manilkara Zapota L in treating bacterial infection"

_PeerJ, doi:10.7717/peerj.17890_

## Round 0.1 · original submission · Major Revisions

As mentioned by R3, Manilkara zapota, like other tropical fruits considered exotic or less commonly consumed, is a tropical fruit that has been little studied, and this is reflected in the brevity of the available literature.

It is important to provide a more in-depth description of the fruit and to utilize some of the available tables to further develop the text.

Reviewer 1 ·

Basic reporting

Dear Editor
I have checked and read the manuscript. My first impression that the paper contains new information and title of the manuscript cover its content. The summary is appropriate and the aim of the work clearly established. The article is scientific and good and is interesting and useful for the reader. Also, It is well written and is scientifically very good and coherent and thus acceptable. This work is a very interesting study and I recommend publishing it.
Regards

Experimental design

good

Validity of the findings

good

Reviewer 2 ·

Basic reporting

Please take note of the following points:
1. Figure 1 is not referenced in the manuscript.
2. Certain words need to be italicized (line 82, table 2).
3. The table should specify whether a qualitative or quantitative composition is being reported. Standardization is needed.
4. In Table 2, it should be made clear that Aspergillus is a fungus; the current wording implies that it is a bacterium.
5. Both tables should explain the meanings of the abbreviations.
6. The manuscript contains numerous sentences with repetitive ideas, for example lines 122, 140, 146, 233.

Experimental design

The review of M. zapota is currently relevant and may be useful for treating antibiotic-resistant bacterial infections. However, it lacks a clear description of the methodology for ensuring the reproducibility of results. For instance, the number of articles found, included, and excluded is not specified. It would be helpful to include the database of articles as additional files. Furthermore, the search terms and Boolean operators used are not clearly explained.

Validity of the findings

Please consider the following question: the conclusions are summarized (line 255).
The manuscript focuses on the potential use of phytochemicals from M. zapota against resistant bacteria. However, it does not discuss or elaborate on this topic or cover research on resistant strains.

Additional comments

I suggest the following to enhance the review:
1) Provide details of the literature search results including the number of articles found, duplicate articles, excluded articles, and included articles.
2) Limit the search to the past 10 years.
3) Include phytochemical sections for each plant part: leaf, stem, fruit, seed, and root.
4) Focus the search and discussion on resistant bacteria, aligning with the orientation of the work.
5) Justify how this review differs from others, such as the review by Priyanka, S., Aakash, D., Harish, K., Nitin, B., Sanjiv, K., & Davinder, K. (2024) on the pharmacological potential of Manilkara zapota (L.) P. Royen (Sapodilla) published in the Journal of Traditional Chinese Medicine, 44(2), 4.
6) Deepen the discussion on how these phytochemicals inhibit bacteria and why they can prevent bacterial resistance.

Reviewer 3 ·

Basic reporting

It is necessary to check the spelling of the scientific names of both fruits and microorganisms as they are written in italics.
There are very repetitive phrases such as M. zapota showed significant antibacterial properties owing to the presence of diverse phytochemical, compounds, such as flavonoids and alkaloids,...

Line 32. Phototerapy??
when describing the different ways of naming malinkara zapota, he does not mention manila sapodilla or chicozapote,
perhaps it would be worthwhile to annex them, however, it is the author's criterion

Experimental design

The manuscript is very short for a review. Perhaps it should take the elements of a table and develop them in greater depth.
In fact, the writing should be developed in more depth.

Validity of the findings

No comments

Additional comments

Manilkara zapota like other tropical fruits considered exotic or little consumed is a tropical fruit little studied and this is reflected in the shortness of the review. It could be considered a mini-review or, alternatively, describe more deeply what is written and use some of the tables for further development of the text.

---

## Round 0.2 · Minor Revisions

The authors mention a search result in the databases of more than 2000 articles screened. However, there is no substantial evidence. Please include a list of the articles that resulted from their search as supplementary data. Also, consider all the minor comments from the reviewers in the revised version.

Reviewer 1 ·

Basic reporting

good

Experimental design

good

Validity of the findings

good

Additional comments

Dear Editor
The authors have revised the manuscript appropriately and have addressed all of the issues raised by the reviewers, making the manuscript acceptable for publication.
Regards

Reviewer 2 ·

Basic reporting

Comments from the first revision have been resolved. Some words needs to be italicized, review the document again.

Experimental design

The diagram for the item selection method figure is incomplete; it should include the final number of papers included.

Validity of the findings

No comments in this second revision.

Additional comments

Boolean operators should be capitalized. Could you include evidence of the results of your database search? You may attach them as additional material.

Reviewer 3 ·

Basic reporting

Please correct on line 33. the word phototherapy..in the conclusion made change to antimicrobial.

Experimental design

No comments

Validity of the findings

No comments

Additional comments

No comments

---

## Round 0.3 · Minor Revisions

I have made a small number of editorial suggestions to improve the manuscript. Please review the attached PDF file and consider these suggestions in the final revised version.

---

## Round 0.4 · accepted · Accept

Thank you for addressing all technical and editorial suggestions for improvement. Your manuscript has been accepted in PeerJ.